# RTGen: Real-Time Generative Detection Transformer

## Abstract

Although open-vocabulary object detectors can generalize to unseen categories, they still rely on predefined textual prompts or classifier heads during inference. Recent generative object detectors address this limitation by coupling an autoregressive language model with a detector backbone, enabling direct category name generation for each detected object. However, this straightforward design introduces structural redundancy and substantial latency. In this paper, we propose a **R**eal-**T**ime **GEN**erative Detection Transformer (RTGen), a real-time generative object detector with a succinct encoder-decoder architecture. Specifically, we introduce a novel Region–Language Decoder (RL-Decoder) that jointly decodes visual and textual representations within a unified framework. The textual side is organized as a Directed Acyclic Graph (DAG), enabling non-autoregressive category naming. Benefiting from these designs, RTGen-R34 achieves 131.3 FPS on T4 GPUs, over $270\times$ faster than GenerateU. Moreover, our models learn to generate category names directly from detection labels, without relying on external supervision such as CLIP or pretrained language models, achieving efficient and flexible open-ended detection.

## 1 Introduction

Object detection (Ren et al., 2016; Lin et al., 2017; Redmon et al., 2016; Carion et al., 2020) has traditionally been limited to a closed set of predefined categories, which motivated the development of open-vocabulary detection (OVD) techniques. Despite remarkable advances (Li et al., 2022b; Gu et al., 2021; Zhou et al., 2022; Minderer et al., 2022), existing OVD models still depend on a limited set of predefined textual prompts during inference, constraining their scalability and adaptability in real-world scenarios. To address this limitation, generative object detection (Yao et al., 2024; Lin et al., 2024) has recently emerged as a promising paradigm that enables open-ended category generation—allowing detectors to produce free-form textual descriptions beyond fixed categories.

Several approaches (Wu et al., 2025; Lin et al., 2024; Yao et al., 2024; Johnson et al., 2016) have explored this direction, though often employing relatively simple designs. For example, GenerateU (Lin et al., 2024) feeds object queries from Deformable DETR (Zhu et al., 2020) into an autoregressive language model to generate object names, while DetCLIPv3 (Yao et al., 2024) attaches an object captioner to an open-vocabulary detector to produce region-level captions. In general, these methods transmit object features extracted by a detector into a separate language model to generate textual outputs. Although effective, such decoupled architectures exhibit weak integration between detection and language generation, leading to redundant computation and limited cross-modal synergy.

Inspired by extensive multimodal research (Li et al., 2023; Bao et al., 2022; Li et al., 2022a; Lu et al., 2019) demonstrating the effectiveness of shared transformer architectures for unified vision–language understanding, we propose a Region–Language Decoder (RL-Decoder) that integrates object detection and text generation within a single decoding framework. Unlike previous generative detectors that attach an external language model, our design treats object queries and text embeddings as co-evolving representations processed jointly through cross-modal attention. This unified formulation allows the model to dynamically align visual and textual features at each decoding layer, thereby enhancing cross-modal interaction and eliminating redundant computation between separate modules.

Figure 1: Comparison of object detection paradigms.

Figure 2: Structure of the proposed DAG Text Head. It estimates token transition probabilities via the Link Prediction Module and constructs a directed acyclic graph for non-autoregressive text generation.

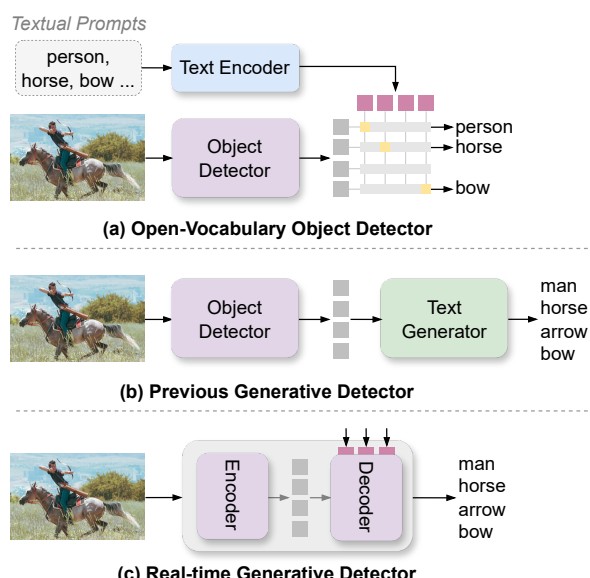

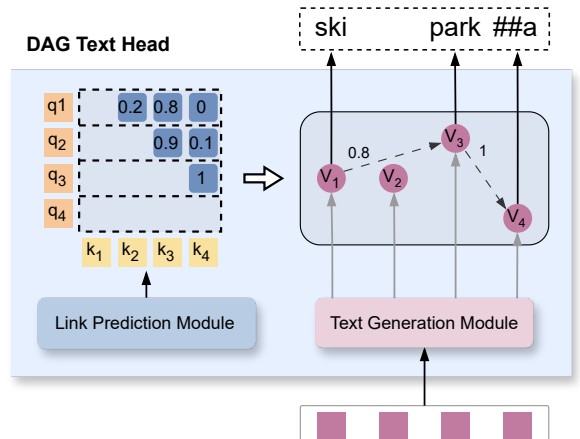

Nevertheless, existing language modeling paradigms are predominantly autoregressive (Radford et al., 2019; Touvron et al., 2023; Chowdhery et al., 2023), which rely on sequential decoding and ground-truth supervision—both incompatible with concurrent reasoning and real-time inference. To address this, we reformulate text generation within the RL-Decoder as a non-autoregressive Directed Acyclic Graph (DAG) generation process, inspired by (Gu et al., 2017). By structuring text embeddings as a DAG to capture token dependencies in parallel, our approach enables simultaneous reasoning over objects and textual concepts, achieving efficient generation without requiring pre-aligned textual inputs.

Building upon the unified RL-Decoder and the DAG-based non-autoregressive generation mechanism, we introduce the Real-Time GENerative Detection Transformer (RTGen), a unified and efficient framework for open-ended object detection. As illustrated in Fig. 1, unlike open-vocabulary detectors (Cheng et al., 2024; Li et al., 2022b; Liu et al., 2025; Zhang et al., 2022) that rely on predefined category prompts or previous generative detectors (Wu et al., 2025; Lin et al., 2024; Yao et al., 2024; Long et al., 2023) that generate text only after detection, RTGen performs joint detection and text generation within a single decoding process. Built upon the efficient RT-DETR architecture (Zhao et al., 2024), our model incorporates the proposed Region–Language Decoder (RL-Decoder) and a non-autoregressive DAG-based generation mechanism, enabling concurrent reasoning over visual and textual representations for efficient and open-ended detection.

In summary, our main contributions are as follows:

- We propose the Real-Time GENerative Detection Transformer (RTGen), a unified and efficient framework that achieves real-time open-ended detection without relying on predefined categories, making it suitable for practical deployment.

- We design a novel Region–Language Decoder (RL-Decoder) that enables concurrent reasoning over visual and textual representations within a single framework, and further produces category names in a non-autoregressive manner for efficient open-ended generation.

- Through these designs, RTGen-R34 reaches 131.3 FPS on T4 GPUs with TensorRT FP16, compared with 0.48 FPS on GenerateU, and RTGen-R101 achieves 40.7 AP on COCO, without relying on CLIP or any pretrained language models.

Figure 3: Overall architecture of RTGen. RTGen builds upon RT-DETR by introducing a unified Region–Language Decoder (RL-Decoder) that jointly processes object queries and positional text embeddings. The refined queries and text features are sent to a detection head and the DAG Text Head, enabling efficient real-time open-ended detection.

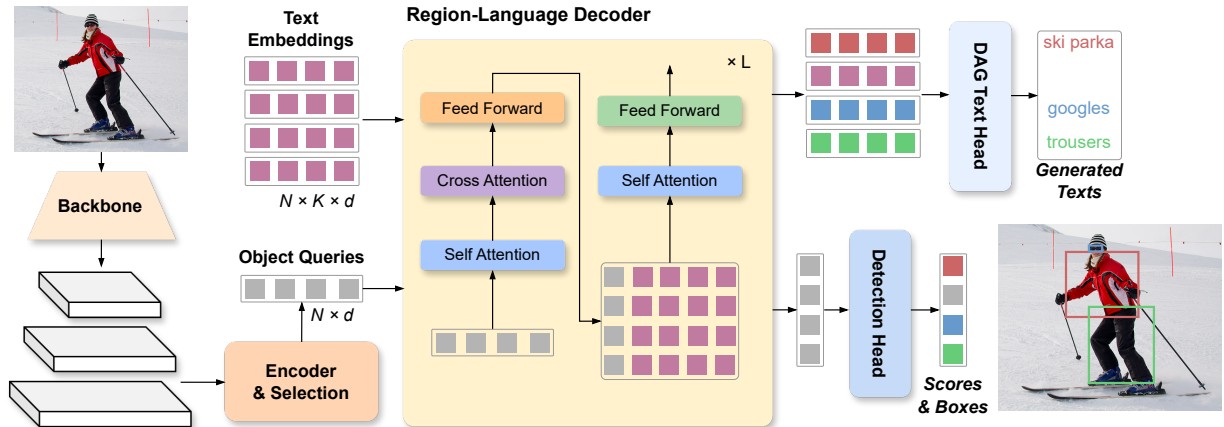

## 2 Method

In this section, we present RTGen, a real-time generative object detection transformer. To achieve efficient integration of detection and language generation, we propose a Region–Language Decoder (RL-Decoder) that concurrently processes object queries and textual representations within a unified decoding structure. Furthermore, the decoder incorporates a DAG-based non-autoregressive text generation mechanism, inspired by (Huang et al., 2022), enabling parallel generation of category names alongside object decoding. The overall architecture of RTGen is illustrated in Fig. 3. We describe the model architecture in Sec. 2.1, the RL-Decoder in Sec. 2.2, the DAG Text Head in Sec. 2.3, and the training formulation in Sec. 2.4.

### 2.1 Model Architecture

We adopt RT-DETR (Zhao et al., 2024), a closed-set object detector, as the foundation of our architecture. As a transformer-based detector derived from DETR (Carion et al., 2020), it employs an encoder–decoder design rather than a conventional CNN backbone, offering stronger compatibility for integration with language models. Moreover, RT-DETR achieves an excellent balance between accuracy and efficiency, outperforming both traditional DETR and Deformable-DETR (Zhu et al., 2020) while maintaining real-time performance. It incorporates an efficient hybrid encoder that disentangles intra-scale interactions and cross-scale integration among different features, as well as a decoder with deformable attention (Zhu et al., 2020) for content-aware sampling. In this work, we disable its classification functionality and adopt a class-agnostic detection setting, focusing solely on object detection and localization.

The overall architecture of our proposed RTGen is illustrated in Fig. 3. Built upon RT-DETR (Zhao et al., 2024), RTGen inherits its real-time efficiency from the retained backbone and encoder, while introducing a novel decoding structure and a text generation head to enable open-ended object detection. The architecture consists of a backbone, an encoder, a Region–Language Decoder (RL-Decoder), a DAG Text Head, and a detection head. The backbone extracts hierarchical visual representations from input images, and the encoder transforms them into context-aware object queries. These object queries are jointly decoded with positional text embeddings in the RL-Decoder, which concurrently enhances object semantics and builds corresponding textual representations, achieving unified reasoning across visual and textual modalities. The updated object queries are sent to the detection head to predict objectness scores and bounding boxes, whereas the refined text embeddings are passed to the DAG Text Head, which organizes them into a directed acyclic graph,

---

**Algorithm 1** Region-Language Decoder Forward Process

---

**Input:** object queries $\mathbf{Q} \in \mathbb{R}^{N \times d}$, image features $\mathbf{M} \in \mathbb{R}^{M \times d}$, language embeddings $\mathbf{T} \in \mathbb{R}^{N \times K \times d}$, number of layers $L$

**Output:** Updated queries $\mathbf{Q}_{L+1}$ and text features $\mathbf{T}_{L+1}$

1: **Initialize: $\mathbf{Q}_1 \leftarrow \mathbf{Q}$, $\mathbf{T}_1 \leftarrow \mathbf{T}$**
2: **for** $l = 1$ to $L$ **do**
3:     **Region-aware update:**

$$\mathbf{Q}_{l+1} = \mathrm{FFN}_r(\mathrm{CrossAttn}(\mathrm{SelfAttn}(\mathbf{Q}_l), \mathbf{M})) \tag{1}$$

4:     **Language fusion:**

$$\mathbf{H}_l = \mathrm{Concat}_{\mathrm{dim}=2}(\mathbf{Q}_{l+1}, \mathbf{T}_l)^{\top (1,2)} \tag{2}$$

5:     **Cross-modal interaction:**

$$\tilde{\mathbf{H}}_l = \mathrm{FFN}_c(\mathrm{SelfAttn}(\mathbf{H}_l)) \tag{3}$$

6:     **Feature separation:**

$$\_, \mathbf{T}_{l+1} = \mathrm{Split}\left(\tilde{\mathbf{H}}_l^{\top (1,2)}, \mathrm{dim}=2, \mathrm{size}=[1, K]\right) \tag{4}$$

7: **end for**

---

models token dependencies via attention, and generates category names in a non-autoregressive manner. Through this integrated architecture, RTGen achieves efficient, real-time open-ended detection with unified object understanding and language generation.

The design of RTGen is motivated by a key insight into unified multimodal modeling. Prior works such as VL-BERT (Su et al., 2019), UNITER (Chen et al., 2020), and BLIP-2 (Li et al., 2023) have shown that different modalities can be processed within a shared transformer framework. However, existing generative object detectors typically adopt a cascaded design, where object decoding and category name generation are handled by separate decoders, resulting in redundant processing of visual information. In contrast, following the unified architecture of RT-DETR described above, we argue that a single decoding stage is sufficient for both tasks. To achieve this, we reformulate category name generation as a non-autoregressive process, enabling text embeddings to be processed in parallel with object queries within one decoder pass. This leads to our Region–Language Decoder, which jointly refines object queries and injects region-specific semantics into text embeddings, enabling efficient and unified cross-modal reasoning.

### 2.2 RL-Decoder

The decoder serves as the core component for cross-modal fusion, where object queries propagate visual semantics into text embeddings. To begin the decoding process, we initialize the positional and semantic components of the text input. Each token slot is associated with a learned positional embedding to provide a distinct identity, while the semantic component is initialized with a special padding token from the CLIP vocabulary. This content-agnostic placeholder—analogous to masked tokens in masked language modeling—allows the decoder to iteratively refine these representations into meaningful category predictions.

As detailed in Algorithm 1, each decoder layer refines object queries and fuses them with text embeddings through four sequential operations. First, a region-aware update is performed, where the object queries $\mathbf{Q}_l$ are successively processed by self-attention, cross-attention with image features $\mathbf{M}$, and a feed-forward network $\mathrm{FFN}_r$, as formulated in Eq. 1. Next, the refined queries $\mathbf{Q}_{l+1}$ are concatenated with the text embeddings $\mathbf{T}_l$ to form a joint representation $\mathbf{H}_l$, as shown in Eq. 2. Cross-modal interaction is then applied via self-attention and another feed-forward network $\mathrm{FFN}_c$ to propagate region-level semantics into the text embeddings, as described in Eq. 3. Finally, the fused representation $\tilde{\mathbf{H}}_l$ is transposed and split along the token dimension to obtain the updated text embeddings $\mathbf{T}_{l+1}$, as shown in Eq. 4.

Notably, we employ a shared self-attention module across both the region-aware and cross-modal stages. This design follows the well-established architectural paradigm in vision-language modeling—as seen in prior works such as VL-BERT (Su et al., 2019), UNITER (Chen et al., 2020), and BLIP-2 (Li et al., 2023)—where a unified Transformer-based mechanism is used to jointly process multi-modal tokens. From a conceptual perspective, self-attention is inherently modality-agnostic; by projecting visual and textual features into a shared latent space, the model is encouraged to learn unified semantic representations while maintaining parameter efficiency. Throughout this process, only the queries updated from the region-aware step are propagated to the next layer, ensuring that textual features do not interfere with the core object queries.

## 2.3 DAG Text Head

After aggregating information from the object queries in the decoder, the text embeddings are passed into the DAG Text Head, which adopts a DAG-based non-autoregressive text generation approach inspired by (Huang et al., 2022), enabling parallel category name generation. Specifically, the output text embeddings from the decoder, denoted as $\mathbf{T}_{L+1}$, are decomposed into $N$ components $\{\mathbf{T}_{L+1}^{(n)}\}_{n=1}^{N}$, as shown in Eq. 5, where $L$ indicates the layer index of the decoder. Each component $\mathbf{T}_{L+1}^{(n)} \in \mathbb{R}^{K \times d}$ represents a sequence of $K$ token embeddings associated with the $n$-th detected region. For each $\mathbf{T}_{L+1}^{(n)}$, query and key representations are obtained through linear projections using learnable matrices $\mathbf{W}_{\mathrm{Q}}$ and $\mathbf{W}_{\mathrm{K}}$, respectively, as defined in Eq. 6. The transition probability matrix $\mathbf{E}$ of the directed acyclic graph is then computed by applying the scaled dot-product attention mechanism, as formalized in Eq. 7. This matrix captures the directed dependencies among tokens, allowing the DAG Text Head to efficiently model non-autoregressive text generation across all region-specific text embeddings in parallel.

$$\mathbf{T}_{L+1} \in \mathbb{R}^{N \times K \times d} \quad \Rightarrow \quad \left\{ \mathbf{T}_{L+1}^{(n)} \in \mathbb{R}^{K \times d} \right\}_{n=1}^{N} \tag{5}$$

$$\mathbf{Q} = \mathbf{T}_{L+1}^{(n)} \mathbf{W}_{\mathrm{Q}}, \quad \mathbf{K} = \mathbf{T}_{L+1}^{(n)} \mathbf{W}_{\mathrm{K}}. \tag{6}$$

$$\mathbf{E} = \mathrm{softmax}\left(\frac{\mathbf{Q}\mathbf{K}^{\top}}{\sqrt{d}}\right). \tag{7}$$

A key advantage of this DAG formulation is that it eliminates the need for autoregressive teacher-forcing during training. Instead of conditioning on ground-truth token sequences, the DAG Text Head learns category-level text representations directly from the matched ground-truth annotations during detection training. Specifically, object queries are first aligned with ground-truth boxes using Hungarian matching, and the corresponding category labels serve as supervision signals for learning token dependencies through the attention-based transition matrix. This formulation enables efficient parallel text generation while remaining consistent with the detection training pipeline.

Among the decoding options for the DAG Text Head, Viterbi decoding (Shao et al., 2022) provides a principled way to recover globally coherent sequences via dynamic programming. Unlike greedy or lookahead decoding, which focus on local or partial likelihoods, Viterbi jointly evaluates all transitions to yield deterministic and globally optimal outputs. Given its superior stability and efficiency over sampling-based approaches and beam search, all subsequent experiments use Viterbi decoding as the default strategy.

## 2.4 Training Formulation

**Training Data.** In traditional object detection, the training sample is defined as $(x, \{\mathbf{b}_i, c_i\}_{i=1}^{N})$, where $x \in \mathbb{R}^{3 \times H \times W}$ is the input image, $\{\mathbf{b}_i | \mathbf{b}_i \in \mathbb{R}^4\}_{i=1}^{N}$ denotes the bounding boxes, and $\{c_i\}_{i=1}^{N}$ are discrete category labels. In contrast, we reformulate the data as $(x, \{\mathbf{b}_i, y_i\}_{i=1}^{N})$, where $\{y_i\}_{i=1}^{N}$ denotes the category name in text form. Unlike open-vocabulary detectors that require predefined category names as input queries, RTGen directly takes the image $x$ and learns to predict both boxes $\{\hat{\mathbf{b}}_j\}_{j=1}^{K}$ and the corresponding generated texts $\{\hat{y}_j\}_{j=1}^{K}$.

Table 1: Comparison of closed-set, open-set, and open-ended object detection methods on the COCO validation set (Lin et al., 2014). RTGen achieves competitive accuracy while significantly outperforming prior open-ended models in speed. The FPS is measured on an NVIDIA T4 GPU using TensorRT FP16 optimization, with an input size of (640, 640). Dataset abbreviations: OI denotes OpenImages (Krasin et al., 2017), O365 denotes Objects365 V1 (Shao et al., 2019), and VG denotes Visual Genome (Krishna et al., 2017).

| Method | BackBone | Type | Supervision | Training Data | #Params (M) | GFLOPs | FPS | AP |
|---|---|---|---|---|---|---|---|---|
| Faster R-CNN (Girshick, 2015) | R50 | Closed-Set | - | COCO | 42 | 180 | - | 40.2 |
| Faster R-CNN (Girshick, 2015) | R101 | Closed-Set | - | COCO | 60 | 246 | - | 42 |
| DETR-DC5 (Carion et al., 2020) | RN50 | Closed-Set | - | COCO | 41 | 187 | - | 43.3 |
| DETR-DC5 (Carion et al., 2020) | R101 | Closed-Set | - | COCO | 60 | 253 | - | 44.9 |
| OWL-ViT (Minderer et al., 2022) | ViT-B | Open-Set | CLIP | OI, VG | - | - | - | 30.3 |
| OWL-ViT (Minderer et al., 2022) | ViT-L | Open-Set | CLIP | OI, VG | - | - | - | 34.7 |
| ViLD (Gu et al., 2021) | R50 | Open-Set | CLIP | LVIS base | - | - | - | 36.6 |
| OV-DETR (Zang et al., 2022) | R50 | Open-Set | CLIP | LVIS base | - | - | - | 38.1 |
| GenerateU (Lin et al., 2024) | Swin-T | Open-Ended | FlanT5-base | VG | 297 | - | 0.48 | 33.0 |
| GenerateU (Lin et al., 2024) | Swin-T | Open-Ended | FlanT5-base | VG, GRIT | 297 | - | 0.48 | **33.6** |
| RTGen | R50 | Open-Ended | - | O365 | 71 | 225 | **90.4** | 32.0 |
| RTGen | R101 | Open-Ended | - | O365 | 105 | 348 | 59.7 | **33.6** |
| RTGen | R50 | Open-Ended | - | O365+COCO | 71 | 225 | **90.4** | 39.7 |
| RTGen | R101 | Open-Ended | - | O365+COCO | 105 | 348 | 59.7 | **40.7** |

**Training Objective.** Given the supervision $\{\mathbf{b}_i, y_i\}$, RTGen jointly optimizes localization and text generation through a unified objective. For each matched prediction $(\hat{\mathbf{b}}_j, \hat{y}_j)$, we apply a box regression loss $\mathcal{L}_{\text{reg}}$, an IoU-based localization loss $\mathcal{L}_{\text{iou}}$, a DAG text generation loss $\mathcal{L}_{\text{DAG}}$, and an objectness loss $\mathcal{L}_{\text{obj}}$. The Hungarian matching cost uses the predicted objectness score instead of a classification score, consistent with our class-agnostic setting. Furthermore, $\mathcal{L}_{\text{DAG}}$ is dynamically scaled by the IoU between $\hat{\mathbf{b}}_j$ and its matched ground-truth $\mathbf{b}_i$, amplifying high-quality matches while reducing the influence of poorly localized predictions.

## 3 Experiments

In this section, we present a comprehensive evaluation of the proposed RTGen. We first describe the experimental settings, including datasets, the evaluation protocol, and implementation details. Subsequently, we report the performance of RTGen under various metrics and present ablation studies to examine the impact of each component in our framework.

### 3.1 Experimental Settings

**Datasets.** We conduct experiments on two large-scale detection datasets: COCO (Lin et al., 2014) and Objects365 V1 (Shao et al., 2019). COCO contains 80 object categories with diverse scenes and dense annotations, while Objects365 provides 365 categories with significantly larger coverage and richer contextual diversity. These complementary datasets allow us to comprehensively evaluate RTGen, where we only utilize their category names without using the original classification labels.

**Evaluation Protocol.** Evaluating open-ended category generation is challenging, as a single object may correspond to multiple semantically valid names, such as 'couch' and 'sofa'. Direct string matching cannot capture such semantic flexibility. Following (Lin et al., 2024), we measure similarity in a continuous embedding space instead of relying on exact lexical matches. A pretrained text encoder projects both generated names and ground-truth annotations into a shared semantic space; each generated name is matched to the ground-truth category with the highest similarity, and this maximum similarity is used to rescale the model's prediction score. The text encoder is used only for evaluation. In our experiments, we adopt the CLIP text encoder (Radford et al., 2021) for its stable and widely used embedding space.

**Implementation Details.** Following the settings of RT-DETR (Zhao et al., 2024), we train three variants of RTGen with ResNet-34, ResNet-50, and ResNet-101 (He et al., 2016) backbones, where the hidden di-

Table 2: Performance of RTGen variants trained and evaluated on the COCO dataset. We report inference speed (FPS) together with AP, $AP_{50}$, and $AP_{75}$ scores.

| Method | FPS | AP | $AP_{50}$ | $AP_{75}$ |
|---|---|---|---|---|
| RTGen-R34 | 131.3 | 34.8 | 46.7 | 37.6 |
| RTGen-R50 | 90.4 | 38.8 | 51.2 | 42.0 |
| RTGen-R101 | 59.7 | 39.6 | 52.2 | 43.0 |

Table 4: Ablation study on the number of decoder layers.

| Layers | AP | Params | GFLOPs | FPS |
|---|---|---|---|---|
| 5 | 38.2 | 70 | 219 | 92.5 |
| 6 | 38.8 | 71 | 225 | 90.4 |
| 7 | 38.5 | 73 | 232 | 86.5 |
| 8 | 38.2 | 75 | 238 | 83.9 |

Table 3: Zero-shot generative object detection results on COCO. Numbers in gray indicate models trained only on COCO's base categories.

| Method | Type | Novel $AP_{50}$ | Base $AP_{50}$ | Overall $AP_{50}$ |
|---|---|---|---|---|
| OVR-CNN | Open-Set | 22.8 | 46.0 | 39.9 |
| Region-CLIP | Open-Set | 26.8 | 54.8 | 47.5 |
| ViLD | Open-Set | 27.6 | 59.5 | 51.3 |
| Detic | Open-Set | 27.8 | 47.1 | 42.0 |
| OV-DETR | Open-Set | 29.4 | 61.0 | 52.7 |
| VLDet | Open-Set | 32.0 | 50.6 | 45.8 |
| RTGen-R50 | Open-Ended | 35.5 | 44.3 | 43.9 |
| RTGen-R101 | Open-Ended | **37.0** | 45.9 | 45.6 |

Figure 4: We report AP, $AP_{50}$, and $AP_{75}$ on the COCO validation set using RTGen-R50 trained on COCO. The results show that using 8 text tokens achieves the best overall performance.

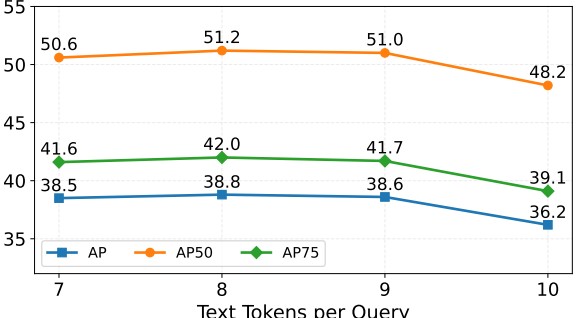

mensions are set to 256, 256, 384, respectively. The decoder consists of 6 layers with 8 attention heads. We select 300 object queries from the encoder features, and each query is associated with 8 text embeddings for DAG-based generation. We use AdamW (Loshchilov & Hutter, 2017) as the optimizer with a base learning rate $1 \times 10^{-4}$, a backbone learning rate $1 \times 10^{-5}$, $\beta$ values of 0.9 and 0.999, and a weight decay of $1 \times 10^{-4}$. All models are trained for 72 epochs on COCO and 12 epochs on Objects365, using a batch size of 16. The overall training objective is formulated as

$$\mathcal{L} = \lambda_{\mathrm{reg}}\mathcal{L}_{\mathrm{reg}} + \lambda_{\mathrm{iou}}\mathcal{L}_{\mathrm{iou}} + \lambda_{\mathrm{obj}}\mathcal{L}_{\mathrm{obj}} + \lambda_{\mathrm{DAG}}\mathcal{L}_{\mathrm{DAG}},$$

where the loss weights are set to $\lambda_{\mathrm{reg}} = 5.0$, $\lambda_{\mathrm{iou}} = 2.0$, $\lambda_{\mathrm{obj}} = 1.0$, and $\lambda_{\mathrm{DAG}} = 1.0$. The DAG loss is computed without normalizing by the number of ground-truth boxes.

## 3.2 Generative Object Detection

Table 1 compares RTGen with representative closed-set, open-set, and open-ended detectors on COCO. Unlike existing open-set and open-ended approaches that rely on massive external semantic supervision, such as CLIP (Radford et al., 2021), trained on hundreds of millions of image–text pairs, or large language models like FlanT5-base (Chung et al., 2024), RTGen does not use any external text or vision–language pre-training. Instead, it learns to produce category names directly from the annotations provided by standard detection datasets, and this generative capability emerges even though the matched training boxes are not perfectly aligned with ground-truth boxes.

With 71 million to 105 million parameters and a computational cost between 225 and 348 GFLOPs, RTGen is substantially lighter than GenerateU, which contains nearly 300 million parameters and requires significantly more computation. This compact design provides a strong efficiency advantage: RTGen-R50 reaches 90.4 FPS on a single T4 GPU using TensorRT FP16, more than 180× faster than GenerateU, while RTGen-R101 also maintains real-time performance. In the zero-shot setting, where the model is trained only on Objects365 and evaluated on COCO, RTGen achieves 32.0 AP for the R50 backbone and 33.6 AP for the R101 backbone,

already comparable to prior open-ended detectors such as GenerateU, whose accuracy ranges from 33.0 to 33.6 AP. When trained with the combined Objects365 and COCO datasets, RTGen further improves to 39.7 AP for R50 and 40.7 AP for R101. These results demonstrate that strong generative object detection performance can be achieved efficiently using only standard detection annotations.

Table 2 summarizes the performance of RTGen variants trained and evaluated on COCO. As the backbone scales from R34 to R101, the models show steady improvements across all metrics, with AP increasing from 34.8 to 38.8 and reaching 39.6, alongside similar gains in $AP_{50}$ and $AP_{75}$. These trends show that larger backbones provide stronger visual features, which in turn lead to better detection quality and category-name generation. At the same time, all variants maintain high efficiency: even the largest R101 model runs at nearly 60 FPS, while the lightweight R34 model exceeds 130 FPS. Overall, these results show that RTGen scales well with backbone size, improving accuracy while still maintaining real-time speed.

In the zero-shot setting, RTGen demonstrates strong generalization to categories that are never exposed during training. Trained solely on Objects365 and evaluated on COCO novel classes, RTGen surpasses representative open-set approaches, as summarized in Table 3. Specifically, RTGen achieves 35.5 $AP_{50}$ with an R50 backbone and 37.0 $AP_{50}$ with R101—substantially higher than prior open-set detectors. These gains are obtained without any external vision–language pre-training, relying only on the category names provided in Objects365. Meanwhile, RTGen maintains competitive performance on COCO base categories and overall metrics. These results highlight the strength of the generative formulation of RTGen in transferring knowledge across datasets and recognizing previously unseen object categories.

### 3.3 Ablation Study

To better understand the design choices of RTGen, we conduct a series of ablation studies focusing on two key architectural factors: the number of text tokens used for category-name generation and the depth of the decoder. All experiments are performed using the RTGen-R50 model trained and evaluated on the COCO dataset.

**Effect of Text Tokens per Query.** To evaluate how the number of text tokens affects generative prediction quality, we vary the number of text tokens from 7 to 10 and report the corresponding AP metrics (AP, AP50, AP75) on COCO. As shown in Fig. 4, the model achieves the best performance when using 8 tokens, reaching 38.8 AP, 51.2 AP50, and 42.0 AP75. Increasing the token length to 9 yields comparable results, suggesting that moderately long category descriptions are beneficial. However, further expanding the token budget to 10 tokens leads to a noticeable degradation, with AP dropping to 36.2, suggesting that overly long text sequences make it difficult for the object queries in the RL-Decoder to effectively propagate semantic information. These results demonstrate that a compact and well-structured description (8 tokens) provides the best balance between expressiveness and discriminability, enabling RTGen to generate category names that are both informative and tightly grounded to object regions.

**Effect of Decoder Depth.** While a 6-layer decoder is known to be optimal in the original RT-DETR architecture, the RL-Decoder in RTGen adopts a different structure that simultaneously refines object queries and processes text-embedding features. This dual responsibility suggests that increasing the decoder depth may provide additional capacity for text-conditioned reasoning, motivating an ablation on the number of layers. As summarized in Table 4, the 6-layer configuration again delivers the strongest accuracy, achieving 38.8 AP. Reducing the depth leads to a modest drop in performance, whereas further increasing it offers no improvement. Instead, deeper decoders introduce notable overhead: GFLOPs rise from 219 to 238, and FPS decreases from 92.5 to 83.9 when expanding from 5 to 8 layers. Overall, these results indicate that six layers strike an effective balance between accuracy and efficiency for jointly refining object queries and integrating text features, and that additional depth yields diminishing returns.

### 3.4 Visualization

In Fig. 5, we present qualitative visualization results produced by RTGen-R101. The model is trained on Objects365 and applied in a zero-shot setting to COCO val images, without using any COCO annotations. These examples highlight RTGen's ability to generalize to novel object categories and previously unseen

Figure 5: Visualization results from RTGen-R101, trained on Objects365 and inferred in a zero-shot setting on the COCO val.

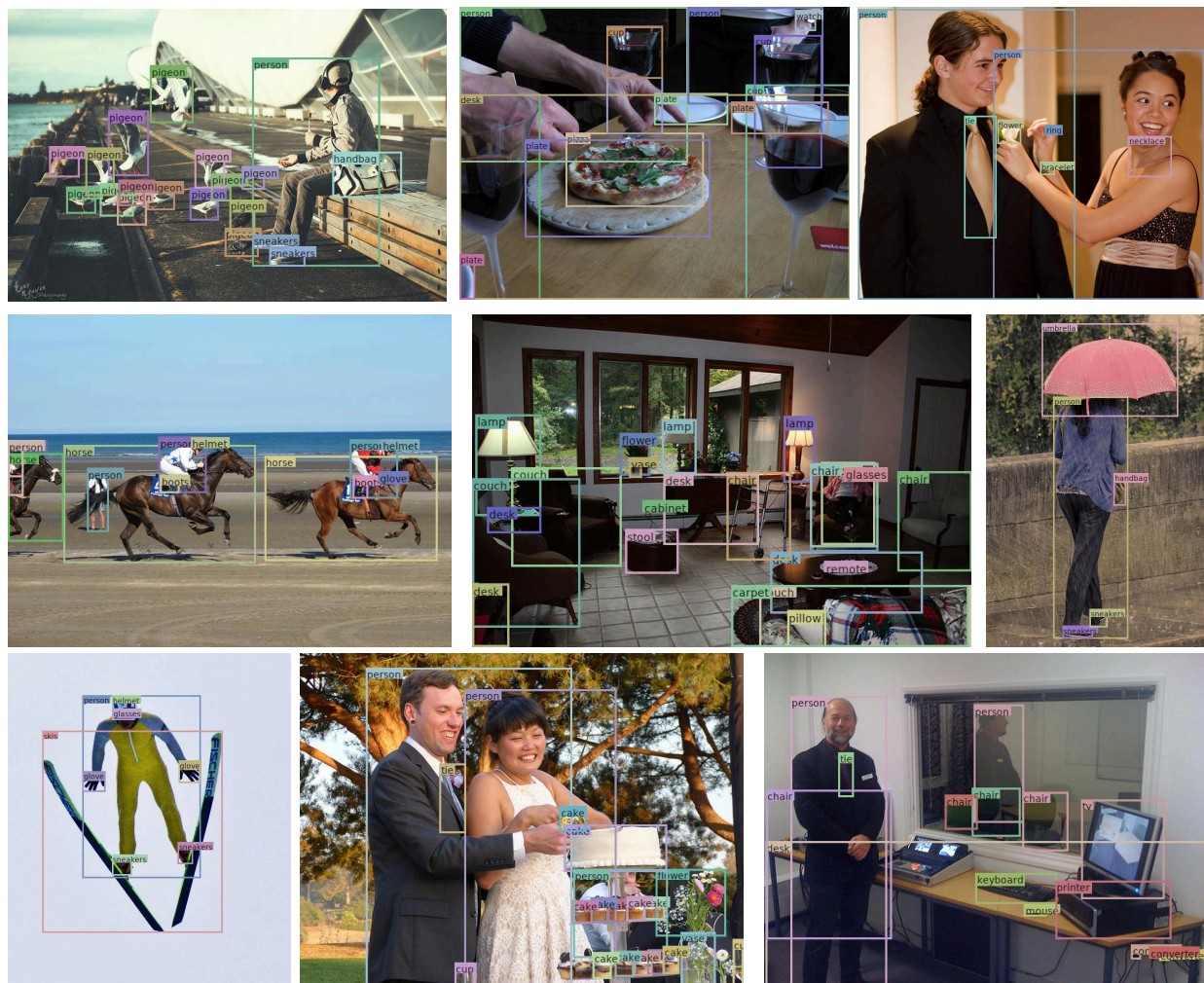

visual concepts, reflecting its open-ended generative formulation. Across diverse scenes, the model produces semantically coherent predictions that align with the visual content, suggesting effective transfer beyond fixed category sets.

We further observe that RTGen maintains consistent localization across varying object scales, cluttered backgrounds, and complex scene compositions, including small or partially occluded instances. Despite these strengths, occasional failure cases are observed, such as imprecise boundaries for heavily occluded objects or confusion between visually similar categories. Overall, these qualitative results complement the quantitative zero-shot benchmarks and provide additional evidence that RTGen supports flexible, open-ended object detection in realistic visual environments.

## 3.5   Limitation

Although RTGen is trained without any external linguistic priors, its vocabulary is inherently limited by the annotations of detection datasets. For example, Objects365 contains only 365 category names, and current detection datasets generally lack large-scale language diversity. As a consequence, the model's generative capacity is constrained, making it unsuitable for zero-shot evaluation on benchmarks such as LVIS

that require a much broader vocabulary, where the dataset vocabulary becomes the dominant bottleneck. Nevertheless, a model trained solely on Objects365 can still generalize reasonably well: when evaluated zero-shot on COCO, RTGen achieves 33.6 AP. This suggests that, given richer linguistic supervision in future detection datasets, RTGen has strong potential to generalize its learned vocabulary to broader domains.

# 4 Related Works

## 4.1 Open-Vocabulary Object Detection

Open-vocabulary object detection (OVD), which aims to recognize objects beyond a fixed set of predefined categories, has recently received increasing attention. Most OVD methods (Li et al., 2022b; Minderer et al., 2022; Cheng et al., 2024; Liu et al., 2025) integrate a pre-trained language encoder into the detector and classify regions by measuring visual–textual similarity. A parallel line of work (Zareian et al., 2021; Zhong et al., 2022; Gu et al., 2021) distills semantic knowledge from vision–language models (VLMs) to enhance the detector's open-set recognition ability. For example, OVR-CNN (Zareian et al., 2021) is an early attempt that leverages grounded image–caption pre-training to align visual and textual representations, while ViLD (Gu et al., 2021) transfers knowledge from the CLIP model (Radford et al., 2021) to a two-stage detector via feature-level distillation. Another strategy for improving OVD performance is to expand the training data through vision–language pre-training. GLIP (Li et al., 2022b) unifies object detection and phrase grounding into a single pre-training framework, and Grounding DINO (Liu et al., 2025) further integrates grounded pre-training into a detection transformer, using multimodal fusion modules to strengthen visual–textual interaction.

Despite significant progress, existing OVD models still require human-provided category names at inference time. This dependence on predefined prompts fundamentally restricts their scalability, adaptability, and applicability in truly open-world scenarios.

## 4.2 Dense Captioning

Dense captioning aims to generate detailed descriptions for specific areas of an image. The task was introduced by Johnson et al. through FCLN (Johnson et al., 2016), a fully convolutional localization network that extracts region proposals with a CNN and localization layer, processes them with a recognition network, and decodes captions using an RNN. CapDet (Joseph et al., 2021) extends this idea to open-world scenarios, feeding unlabeled object proposals into a captioning head to produce free-form descriptions. More recent approaches, such as GRiT (Wu et al., 2025) and DetCLIPv3 (Yao et al., 2024), attach a text generator to an object detector to produce object descriptions. While dense captioning focuses on generating rich, descriptive captions for objects or regions, our setting instead emphasizes predicting concise, canonical category names. Dense captioning is typically evaluated using both AP from object detection and METEOR (Banerjee & Lavie, 2005) from machine translation.

## 4.3 Non-autoregressive Translation

Non-autoregressive translation (NAT) models aim to generate sequences in parallel without conditioning on previously produced tokens, greatly accelerating inference. The idea was introduced by Gu et al. (Gu et al., 2017), though the removal of sequential dependencies leads to a clear accuracy gap relative to autoregressive (AT) models. Subsequent approaches seek to balance speed and dependency modeling: SAT (Wang et al., 2018) adds limited autoregressive steps, CMLM (Ghazvininejad et al., 2019) iteratively refines masked tokens, and DA-Transformer (Huang et al., 2022) uses a directed acyclic graph (DAG) to organize token generation in a fully NAT manner, enabling parallel decoding while preserving structural dependencies. Inspired by this DAG formulation, RTGen adopts a lightweight DAG-based text head that enables efficient non-autoregressive category-name generation, preserving both consistency and dependency awareness among output tokens.

## 5    Conclusion

In this work, we presented RTGen, a real-time generative object detector that integrates category-name generation into real-time detection. Unlike prior open-set or open-ended models that rely on large-scale vision–language pre-training or heavyweight language models, RTGen achieves generative detection using only standard detection annotations. Central to our design is the RL-Decoder, which refines object queries while processing text embeddings, together with a DAG-based generative head that provides fast and consistent category-name decoding. Comprehensive experiments on COCO demonstrate that RTGen delivers competitive accuracy, zero-shot generalization, and substantially higher speed than existing open-ended approaches. Together, these results establish real-time generative detection as a practical and scalable paradigm for open-ended object detection.

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
