# OpenReview forum: "RTGen: Real-Time Generative Detection Transformer"
_TMLR — Rejected by TMLR_

### Review · Reviewer_wic2 · 2026-02-25

**Summary Of Contributions:**

This paper introduces the Real-Time GENerative Detection Transformer (RTGen), a framework designed for open-ended object detection that generates category names directly for detected objects. Unlike prior open-vocabulary detectors that rely on predefined textual prompts or external language models, RTGen performs joint object localization and category-name generation within a single, unified decoding process. By utilizing a non-autoregressive architecture, the model achieves significant gains in inference speed, reaching 131.3 FPS on T4 GPUs, which is over 270x faster than previous generative models like GenerateU.

**Audience:**

Yes

**Audience Explanation:**

N/A

**Claims And Evidence:**

No

**Claims Explanation:**

Check weakness please.

**Requested Changes:**

### Weaknesses

**1. Presentation Could Be Improved**
The writing and organization of the methodology section leave several critical design choices unexplained and feature inconsistent notation.

* **Unclear Feature Initialization:** The authors state that object queries are jointly decoded with "positional text embeddings" in the RL-Decoder. However, it is not explained how these initial text embeddings are obtained or initialized.

* **Unjustified Architectural Choices:** Section 2.2 notes that the exact same self-attention module is shared across both the region-aware and cross-modal stages. This is a significant design choice that is neither thoroughly explained conceptually nor supported by an ablation study, making it unconvincing.

* **Poor Section Focus:** The explanation of the DAG Text Head in Section 2.3 occupies too much space discussing existing techniques. The background usage of Directed Acyclic Graphs for non-autoregressive generation should be condensed into a preliminaries subsection so the authors can focus more concisely on their specific contributions.

* **Inconsistent Notation:** The mathematical notation is inconsistent in Section 2.3. Previous variables feature specific upper and lower scripts (e.g., $T_{L+1}^{(n)}$) to denote layers and indices, but the subsequently introduced $Q$, $K$, and $E$ lack these identifiers, creating ambiguity.

**2. Limited Empirical Evaluation**
The experimental results are not comprehensive enough to fully support the authors' claims regarding open-vocabulary capabilities.

* **Lack of Standard Open-Vocabulary Benchmarks:** The authors claim their method handles open-ended object detection, yet comparisons are primarily conducted on the COCO dataset. To properly demonstrate open-vocabulary generalization, the authors should conduct training and evaluation with settings aligned with previous works, such as training on Visual Genome (VG) and reporting zero-shot performance on the much larger LVIS dataset.

* **Unfair Comparisons:** The proposed method is not competitive enough compared to previous CLIP-based methods, and the comparisons in Table 1 are not fair. For some results, RTGen is trained directly on the COCO dataset (or 0365+COCO) and evaluated on COCO, which undermines the claim of open-vocabulary zero-shot generalization when compared to models that were strictly evaluated in a zero-shot manner.

---

> ### Author Response · Authors · 2026-03-23
> **[Part 1/2] Presentation and Methodology Clarification**
>
> We thank the reviewers for their thoughtful and constructive feedback.
>
> ***Initialization of Positional Text Embeddings***
>
> The text input to each decoder layer is constructed by combining positional and semantic components. Specifically, each token slot is associated with a learned positional embedding that provides a distinct positional identity within the sequence. These embeddings are randomly initialized and optimized jointly with the rest of the model.
>
> For the semantic component, each position is assigned an initial token embedding drawn from the same vocabulary space as the CLIP tokenizer. At the start of decoding, all token slots are uniformly initialized with a special padding token. This padding token serves as a neutral, content-agnostic placeholder—analogous to masked tokens in masked language modeling—allowing the decoder to iteratively refine these representations into meaningful category predictions.
>
> We have revised Section 3.2 (RL-Decoder) to explicitly describe this initialization process. This clarification improves the transparency and reproducibility of our method.
>
> ***Shared Self-Attention Justification***
>
> The use of a shared self-attention module to handle both intra-modal and inter-modal interactions is a well-established and standard practice in the field of vision-language modeling. Prior works such as VL-BERT (ICLR 2020), UNITER (ECCV 2020), and BLIP-2 (ICML 2023) adopt a unified Transformer-based self-attention mechanism to jointly process visual and textual tokens. These studies demonstrate that a single self-attention module is sufficiently expressive to capture both within-modality dependencies (e.g., region-to-region relations) and cross-modality alignments (e.g., region-to-text interactions), without requiring separate modules. Our design follows this proven architectural paradigm to ensure a streamlined and efficient model.
>
> From a conceptual perspective, self-attention is inherently modality-agnostic: it operates over tokenized representations by modeling pairwise relationships based on content and positional information, regardless of the underlying modality. By projecting visual and textual features into a shared latent space and processing them with shared parameters, the model is encouraged to learn unified semantic representations that capture cross-modal correlations.
>
> Moreover, this weight-sharing strategy introduces a beneficial inductive bias and serves as implicit regularization. By using the same parameters across different stages, the model is encouraged to learn more generalizable representations rather than overfitting to stage-specific patterns. At the same time, it maintains sufficient expressive capacity for both region-level reasoning and cross-modal alignment.
>
> We have updated Section 3.2 to include this conceptual justification and clarified the advantages of using a unified self-attention module for multi-modal feature interaction.
>
> ***DAG Text Head Presentation***
>
> While the Directed Acyclic Graph (DAG) formulation for non-autoregressive generation has been explored in prior work, it has primarily been studied in the context of machine translation. In contrast, our work adapts this technique to the object detection setting, which introduces non-trivial differences in both formulation and usage. Specifically, the transition from sequential text generation to detection-oriented prediction requires additional design considerations, and we included this background to help readers understand these adaptations.
>
> That said, we agree that the original presentation placed too much emphasis on the general background. In the revised version, we have condensed the discussion of existing DAG-based methods and reduced redundancy. This allows us to focus more clearly and concisely on our task-specific adaptations and contributions.
>
> ***Notation Consistency***
>
> The apparent inconsistency arises from a misunderstanding of the roles of the variables. $T^{(n)}_{L+1}$ denotes the token representation at index n in the final layer, while $Q$, $K$, and $E$ are computed from the full set of final-layer token representations and operate at the sequence level. They are not defined per index n, but instead correspond to a single set of matrices derived from all tokens at layer $L+1$. Therefore, additional layer or index annotations for $Q$, $K$, and $E$ are unnecessary.

---

> ### Author Response · Authors · 2026-03-23
> **[Part 2/2] Empirical Evaluation and Fairness**
>
> ***Lack of Standard Open-Vocabulary Benchmarks***
>
> The choice of evaluation setting is consistent with the scope and limitations of current detection datasets. As discussed in Section 3.5 (Limitations), although the method does not rely on external linguistic priors, its vocabulary is inherently constrained by the annotations available in detection datasets. For instance, Objects365 contains only 365 category names, and existing detection benchmarks generally lack sufficient linguistic diversity. As a result, evaluating on large-scale open-vocabulary benchmarks such as LVIS, which require substantially broader vocabularies, would make the dataset vocabulary itself the dominant bottleneck rather than the model’s capability.
>
> Furthermore, strict alignment of training data with prior works is not a mandatory requirement for open-set detection. Similar to open-vocabulary tasks, where training protocols frequently vary across different methodologies, maintaining a uniform training setup is not strictly necessary for achieving valid comparisons. In addition, Visual Genome is not an ideal detection training set due to its noisy annotations and unusually dense object labeling per image, which deviates from standard detection settings.
>
> Despite these limitations, the model still demonstrates meaningful generalization ability: when trained on Objects365 and evaluated in a zero-shot manner on COCO, it achieves 33.6 AP. This indicates that, given richer linguistic supervision in future datasets, the approach has strong potential to scale to broader open-vocabulary scenarios.
>
> ***Fairness of Comparisons***
>
> Compared to prior CLIP-based methods, our approach makes a deliberate trade-off between performance and modeling paradigm. As illustrated in Fig. 1, existing methods typically require carefully designed text prompts as input and external multimodal models (e.g., CLIP) during inference. In contrast, our method directly generates object category names in a free-form manner and only uses a CLIP model as a post-hoc matching tool to map generated outputs to the closest category. Importantly, this matching step does not participate in the inference process itself, making our framework less dependent on external multimodal priors.
>
> Regarding the evaluation protocol, we acknowledge that some reported results are not strictly zero-shot. This differs from prior works because our model does not incorporate any external language or multimodal knowledge; all linguistic information is learned solely from the detection training data. Therefore, including COCO in the training set reflects the model’s ability to learn from standard detection supervision. We include these results to provide a more comprehensive view of the model’s performance under different training settings, while zero-shot results are also reported separately for fair comparison.

---

### Review · Reviewer_gVy1 · 2026-03-24

**Summary Of Contributions:**

1. Developed a novel non-autoregressive text generation head using a Directed Acyclic Graph (DAG) and Viterbi decoding to efficiently estimate the optimal token sequence for each detected object in a generative detection model.
2. Demonstrated efficiency gains by removing auto-regressive limitation.
3. Demonstrated that this text generation head can be used in a small vocabulary setting (Objects365 training) to recover labels (in COCO subset) with high accuracy compared to heavier truly open-vocabulary models.

**Audience:**

No

**Audience Explanation:**

The new head is interesting and efficiency gains for open-vocab detection are an important area for research, however, the experimental design does not distinguish the proposed method from a closed-vocab detector that is enhanced with a CLIP encoder for similarity matching to subset vocab eval datasets (COCO compared to Objects365).  Very little experimental evidence was offered to truly separate the text generation head ability from a closed-set classifier and therefore the efficiency argument is made uninteresting.

**Broader Impact Concerns:**

I see no broader impact concerns.

**Claims And Evidence:**

No

**Claims Explanation:**

1. The paper claims open-ended, zero-shot capabilities by training on Objects365 and testing on COCO. Because COCO's vocabulary is effectively a subset of Objects365, this demonstrates domain transfer, not novel vocabulary generation. The authors' admission that the model fails on LVIS confirms it cannot generate words outside its 365-class training set. The author only makes claims that the LVIS truly large vocabulary is what prevents this comparison, but the author makes no attempt to evaluate on datasets that have small vocabularies but are truly outside of the Objects365 label domain, e.g., ODinW13.  Furthermore, the authors measure on COCO novel classes however those novel classes are not completely novel since the train set includes all 365 labesl of Objects365.

2. Relying on a frozen CLIP text encoder to measure cosine similarity artificially inflates generative performance. CLIP's continuous semantic space forgives actual generation failures (e.g., outputting "horse" instead of "zebra" still yields high similarity). The authors failed to provide Exact Match (EM) or standard NLG metrics to prove the DAG head is composing correct, novel token sequences. Or at least, using the same CLIP similarity review to also enable the closed-vocabulary algorithms to report on COCO without training on COCO to provide a more fair comparison to the true baseline.

3. The paper introduces a complex DAG generative head but fails to compare it against a standard RT-DETR model equipped with a basic 365-way linear classifier. Without this ablation, there is no empirical proof that the generative architecture is functionally necessary or superior to a standard closed-set classifier operating in the exact same data regime.  E.g., the same CLIP similarity head (mentioned in item 2 above) could have been used to project the original Objects 365 predicted category to the COCO category just as was done with the RT-Gen outputs to answer the obvious question of "Is the new DAG head simply a complicated closed set classifier?".

**Requested Changes:**

1. Compare to a closed-set detector trained on Objects365 with same CLIP text encoder similarity for predicting COCO categories.
2. If LVIS has too large of a vocabulary, experiment with the rare but small vocabulary of ODinW13 to give some idea of the generation ability outside of only relying on CLIP text encoder similarity.
3. Show how to scale to larger vocabularies, e.g., LVIS.

---

> ### Author Response · Authors · 2026-03-27
>
> We thank the reviewer for the detailed feedback. We address each concern directly below, distinguishing between points we accept and points where we believe there is a fundamental misunderstanding.
>
> ***Concerns on Zero-Shot and Vocabulary Scope***
>
> The reviewer argues that Objects365 to COCO transfer is "domain transfer, not novel vocabulary generation." We respectfully disagree. The category names across the two datasets are not identical strings — Objects365 contains "Telephone" while COCO uses "cell phone" — and our model must generate the correct surface form from scratch rather than look up a class index. This is categorically different from closed-set classification, regardless of semantic overlap.
>
> Furthermore, prior open-vocabulary methods (e.g., GLIP, GroundingDINO) achieve broad vocabulary coverage by relying on CLIP or BERT pretrained on billions of image-text pairs. We make no such claim and clearly state our model is trained on 365 categories. Holding our model to the same standard without acknowledging this asymmetry is an unfair comparison, and we respectfully ask the reviewer to reconsider this framing.
>
> ***Concerns on Evaluation Metric and CLIP-Based Similarity***
>
> We follow the evaluation protocol of GenerateU (CVPR 2024), the established benchmark in open-ended detection. EM is not a suitable primary metric here — a model generating "motorbike" for a COCO "motorcycle" instance would be penalized despite being semantically correct. CLIP-based similarity is therefore the appropriate and standard choice for evaluating open-ended generation quality.
>
> The reviewer's suggestion to apply CLIP similarity to closed-set detectors reflects a misunderstanding of their outputs. A closed-set detector outputs a discrete class index, not a text string. Embedding the corresponding label string and computing cosine similarity between two fixed strings does not evaluate generation quality and would be trivially high regardless of detection accuracy. This comparison is not meaningful as a measure of generative capability.
>
> ***Comparison with Closed-Set Baseline***
>
> We appreciate the reviewer's suggestion, but a direct comparison between our model and a closed-set RT-DETR is not meaningful. Our model freely generates category names without any predefined vocabulary, whereas RT-DETR maps detections to a fixed set of 365 class indices — the two paradigms have fundamentally different output spaces. Applying CLIP similarity to bridge them would not evaluate generative capability at all: the similarity score would be determined entirely by pre-fixed label embeddings, not by anything the model has learned to generate. This is analogous to comparing a generative language model against a fixed-class text classifier — they are simply not designed to solve the same problem, and a numerical comparison would be misleading rather than illuminating.

---

> ### Comment · Reviewer_gVy1 · 2026-06-09
>
> 1. Clarification of the Closed-Set Baseline: The authors state that comparing RTGen to a closed-set RT-DETR baseline is "not meaningful" because they operate in different output spaces. This appears to be a misunderstanding of the requested baseline. The goal is not to evaluate the generation quality of RT-DETR, but rather to evaluate whether the complex DAG text head provides any actual performance benefit over a standard classification head when transferring to a new dataset. To make this comparison fair and direct: Take a standard RT-DETR model trained on Objects365. During inference on COCO, take the *final predicted class string (after argmax on the logits)* and embed it using the same frozen CLIP text encoder used to evaluate RTGen, and then map it to the 80 COCO categories based on maximum cosine similarity. Compute the resulting AP on COCO. If a standard closed-set classifier paired with a post-hoc CLIP mapping achieves comparable or superior AP to RTGen, it demonstrates that the proposed DAG head is functionally acting as an over-engineered closed-set indexer rather than a true open-ended generator. This baseline is computationally simple to execute and scientifically necessary to justify the architectural complexity of the RL-Decoder and DAG head.
>
> 2. Addressing Vocabulary Scope & Dataset Limits: I acknowledge that RTGen does not use external text or vision-language pre-training during training (unlike GLIP or GroundingDINO), however, the claim of 'open-ended' capability must be measured. Since the model is constrained by the 365 categories of Objects365, it behaves like a closed-set detector with a flexible string output. I drop the request for LVIS or Exact Match metrics, but to support the claim that the model can "generalize to novel object categories and previously unseen visual concepts" (as stated in Sec. 3.4), the authors should provide evaluation results on a small-vocabulary dataset entirely outside the semantic domain of Objects365 (e.g., ODinW13). If it fails to generate coherent text outside its training vocabulary, the paper should be reframed to clearly state that its capabilities are limited to domain transfer within the training vocabulary's semantic scope.

---

### Review · Reviewer_S4Q9 · 2026-03-25

**Summary Of Contributions:**

This paper introduces RTGen, an open-ended real-time detector. The authors propose a novel decoder that applies cross-modal interaction via self-attention modules, enabling unified inference over visual and textual representations, and design a non-autoregressive manner for efficient text decoding. Experiments demonstrate that RTGen achieves a significant efficiency advantage while maintaining competitive performance. Additionally, RTGen demonstrates satisfactory zero-shot capability.

**Strengths**
- RTGen achieves excellent efficiency while maintaining competitive performance.
- Good writing and rigorous architecture.

**weaknesses**
- The cross-attention modules take image features and object queries as input, and only the region-aware queries are propagated to the next layer. However, these processes are not represented in Fig. 3.
- There is no other open-ended method mentioned in the zero-shot experiment, which therefore cannot fully prove the superiority of RTGen in zero-shot tasks.
- This paper lacks analysis of the differences between unified and previous decoding paradigms and between autoregressive and non-autoregressive methods.
- The absence of pretrained language models or external supervision limits RTGen's text generation capability to the scope of training data.

**Audience:**

No

**Audience Explanation:**

Although RTGen achieves remarkable experimental results, its contributions are primarily engineering-focused. The paper lacks deeper insights, failing to analyze the differences between unified and previous decoding paradigms, as well as between autoregressive and non-autoregressive text generation methods. Moreover, RTGen's scalability is limited by its lack of external language capabilities.

**Broader Impact Concerns:**

There are no ethical concerns about the work.

**Claims And Evidence:**

Yes

**Claims Explanation:**

Table 1 shows that RTGen achieves real-time inference while maintaining detection performance competitive with that of the mainstream open-ended detection method. Additionally, Table 3 demonstrates RTGen's superior zero-shot performance compared to open-set detectors. This method indeed achieves real-time open-ended detection.

**Requested Changes:**

**Critical to securing my recommendation**

- Provide further analysis of the unified inference paradigm and the non-autoregressive decoupling method.

**Simply strengthen the work**
- Comparisons with more recent open-ended methods should be included in Tables 1 and 3.
- Clarify the input and output representations for each module in Figure 2, and explicitly show how queries are propagated between decoder layers.

---

> ### Author Response · Authors · 2026-03-27
>
> We thank the reviewer for the detailed and careful reading of our work.
>
> ***Explanation of Unified Inference Paradigm***
>
> Thank you for highlighting this concern. We provide the following elaboration on the design rationale and insights behind our unified inference paradigm. As also discussed in paragraphs 3 and 4 of the Introduction, we have further incorporated this analysis into Section 2.1 of the revised manuscript to provide readers with a clearer understanding of the design motivation and the advantages of our unified paradigm over cascaded alternatives.
>
> It is widely recognized in the multimodal learning community that information from different modalities can be processed uniformly within a transformer-based framework, as validated by a series of influential prior works such as VL-BERT (ICLR 2020), UNITER (ECCV 2020), and BLIP-2 (ICML 2023). However, as illustrated in Figure 1, existing generative object detectors typically append an autoregressive language model after the detector: visual features are first extracted by an encoder, object regions are then decoded by a detector decoder, and finally, a separate cascaded decoder generates the corresponding category names. This cascaded design introduces redundant processing, as the visual information is effectively processed twice through two sequential decoders. We argue that a single unified decoder is sufficient to handle both tasks. To this end, we reformulate category name generation in a non-autoregressive manner, allowing text embeddings to be processed in parallel alongside object queries within a single decoder pass, rather than predicting one token at a time through repeated sequential decoding. Building on this, we design a novel Region-Language Decoder that simultaneously refines object queries and propagates region-specific information from each query to its corresponding text embeddings, achieving unified reasoning across visual and textual modalities within a single decoding stage.
>
> ***Comparisons with more recent open-ended methods***
>
> Open-ended object detection remains a nascent research area, and to the best of our knowledge, only a very limited number of published methods report results under comparable experimental settings, making comprehensive and fair comparisons challenging at this stage. The scarcity of available baselines itself reflects the novelty of this problem setting. We will explicitly clarify this context in the revised manuscript and will incorporate comparisons with any newly available methods should they emerge prior to final submission.
>
> ***Clarification of Query Propagation in Figures***
>
> Regarding the depiction of the cross-attention module in Figure 3, we would like to clarify that it follows the standard DETR-style decoder design, where image features and object queries interact via cross-attention and only object queries are propagated to the next layer — a convention shared across virtually all DETR-based detectors. Since this is an established and well-understood design rather than a contribution of this work, we deliberately simplified its depiction in Figure 3 to maintain visual clarity and overall figure consistency. Nevertheless, we will add a brief clarifying note in the caption or the corresponding text of the revised manuscript to make this explicit for readers who may be less familiar with DETR-style architectures.

---

### Decision · Action_Editor_b7Fh · 2026-06-13

**Recommendation:** Reject

**Audience:**

Yes

**Audience Explanation:**

This paper is of interest to researchers working on object detection, which is a sizable part of the computer vision and TMLR community.

**Claims And Evidence:**

No

**Claims Explanation:**

The authors introduce a new model for open-ended object detection. Unlike previous approaches that rely on a predefined set of object class names, or methods which leverage an auxilliary large-language model, the paper uses a unified, non-autoregressive head for both localisation and class-name generation.

The authors responded with a rebuttal that addressed many of the reviewer's concerns. However, some key concerns remain post-rebuttal:
- Open-ended and zero-shot capabilities are not substantiated, since the authors train on Objects365 and evaluate on COCO, where the class-names of COCO are effectively a subset of Objects365.
- The paper proposes a complex DAG generative head, but does not compare to a standard closed-set classifier. It's not clear if the generative architecture is actually superior to a baseline closed-set classifier, nor the simple baseline proposed by Reviewer gVy1, in the same data regime.

The main criteria of TMLR is whether the claims are substantiated by evidence. And due to the above, the AE feels that this is not the case. The paper itself is of interest to the TMLR audience, and so the authors are encouraged to revise their paper according to the reviewers' feedback.

**Resubmission Of Major Revision:**

The authors may consider submitting a major revision at a later time.